# Long working hours are associated with a higher risk of non-alcoholic fatty liver disease: A large population-based Korean cohort study

**Yesung Lee** , **Eunchan Mun** , **Soyoung Park, Woncheol Lee** *

Department of Occupational and Environmental Medicine, Kangbuk Samsung Hospital, Sungkyunkwan University School of Medicine, Seoul, Republic of Korea

* doctor.oem@gmail.com

## Abstract

### Background

Non-alcoholic fatty liver disease (NAFLD), a common chronic liver disease, may progress to fibrosis, cirrhosis, hepatocellular carcinoma, and liver failure. But only a few cross-sectional studies have reported an association of NAFLD with working hours. This cohort study further examined the association between working hours and the development of NAFLD.

### Methods

We included 79,048 Korean adults without NAFLD at baseline who underwent a comprehensive health examination and categorized weekly working hours into 35–40, 41–52, 53–60, and >60 hours. NAFLD was defined as the presence of fatty liver, in the absence of excessive alcohol use, as observed by ultrasound.

### Results

During a median follow-up of 6.6 years, 15,095 participants developed new-onset NAFLD (incidence rate, 5.55 per 100 person-years). After adjustment for confounders, the hazard ratios (95% confidence interval) for the development of NAFLD in 41–52, 53–60, and >60 working hours compared with that in 35–40 working hours were 1.07 (1.02–1.13), 1.06 (1.00–1.13), and 1.13 (1.05–1.23), respectively. Furthermore, the association remained significant after confounders were treated as time-varying covariates.

### Conclusion

In this large-scale cohort, long working hours, especially >60 working hours a week, were independently associated with incident NAFLD. Our findings indicate that long working hours are a risk factor for NAFLD.

**Data Availability Statement:** The data are not available to be shared publicly because of ethical restrictions imposed by the Institutional Review Board of Kangbuk Samsung Hospital. For

additional information to request the data, please contact the Institutional Review Board of Kangbuk Samsung Hospital (IRB, Tel. +82-2001-1943; e-mail: irb.kbsmc@samsung.com).

**Funding:** The author(s) received no specific funding for this work.

**Competing interests:** The authors have declared that no competing interests exist.

## Introduction

Non-alcoholic fatty liver disease (NAFLD) is the most common chronic liver disease, and its incidence has increased rapidly, with a global prevalence of 34–46% [1–3]. NAFLD, which ranges from simple benign steatosis to non-alcoholic steatohepatitis, may progress to fibrosis, cirrhosis, portal hypertension, hepatocellular carcinoma and, liver failure [4]. Given the increasing prevalence and hepatic and extrahepatic complications, it is important to identify modifiable risk factors and develop preventive strategies for NAFLD [5].

Overwork is an emerging issue that not only reduces work efficiency but also threatens the health of workers. According to recent statistics from the Organization for Economic Cooperation and Development, South Korea was one of the countries with the highest annual working hours per worker in 2018 [6]. Studies have demonstrated that long working hours affect general health [7] and probably lead to coronary heart disease [8], stroke [9], obesity [10, 11], hypertension [12], diabetes mellitus (DM) [13], and metabolic syndrome [14, 15], which can cause NAFLD.

Generally, NAFLD is recognized as a hepatic component of metabolic syndrome [16], and NAFLD patients tend to be obese and often have increased risk factors of cardiovascular diseases such as hypertension, DM, and dyslipidemia [17]. These findings suggest that overwork might be associated with the pathogenesis of NAFLD. However, studies regarding the relationship between actual working hours and NAFLD are scarce. A few cross-sectional studies have shown an association of NAFLD with long working hours and shift work [18, 19].

The present cohort study aimed to further examine the direct relationship between long working hours and NAFLD. We evaluated the effects of weekly working hours on the incidence of NAFLD in a large-scale cohort of young and middle-aged individuals who participated in a health screening program.

## Materials and methods

### Study population

The Kangbuk Samsung Health Study is a cohort study of South Korean adults aged 18 years and older who underwent a comprehensive annual or biennial health examination at Kangbuk Samsung Hospital Total Healthcare Center in Seoul and Suwon, South Korea [20]. More than 80% of participants were employees of various companies and local governmental organizations and their spouses. In South Korea, the Industrial Safety and Health Law requires annual or biennial health screening examinations of all employees, free of charge. Other examinees voluntarily underwent health checkups at the health care center [21].

The present study included a total of 217,654 participants who underwent the comprehensive health examinations from January 1, 2012, to December 31, 2017, and had undergone at least one other screening exam before December 31, 2018 [22]. During this period, the baseline data were collected at the time of the first health examination visit. We excluded 138,606 participants based on the following criteria (Fig 1): missing data on abdominal ultrasound findings or average working hours per week at baseline; presence of fatty liver or liver cirrhosis on ultrasound; history of malignancy; known liver disease or current use of medications for liver disease; alcohol intake of ≥30 g/day for men or ≥20 g/day for women; positive serologic markers for hepatitis B or C virus; use of steatogenic medications within the past year such as valproate, amiodarone, methotrexate, tamoxifen, or corticosteroids; aged >65 years or <19 years; and working <35 hours per week [5]. Some participants met more than one exclusion criterion. Finally, a total of 79,048 participants were eligible for this study at baseline. We followed the 79,048 Korean adults without NAFLD at baseline annually or biennially. This study was approved by the Institutional Review Board of Kangbuk Samsung Hospital, which

Participants who underwent health examination between January 1, 2012 and December 31, 2017 and had at least one follow-up visit through December 31, 2018 (n=217,654)

Exclusions (n= 138,606): some individuals met more than one criterion for exclusion
- Missing data on abdominal ultrasound or average working hours per week (n=9,457)
- History of malignancy at baseline (n=4,029)
- History of liver disease or medication use for hepatitis at baseline (n=26,186)
- Use of medication for liver cirrhosis or findings of liver cirrhosis on ultrasound at baseline (n=79)
- Alcohol intake of ≥ 30 g/day for men and ≥ 20 g/day for women (n=43,752)
- Use of steatogenic medication at baseline (n=340)
  (Amiodarone, tamoxifen, methotrexate, valproate, and steroid)
- Positive serologic markers for hepatitis B or C at baseline (n=7,248)
- Fatty liver on ultrasound at baseline (n=69,324)
- Aged more than 65 years or less than 19 years (n=478)
- Working less than 35 hours per week (n=46,998)

Participants included in the final analysis (n=79,048)

**Fig 1. Flowchart of study participants.**

exempted the requirement for informed consent because we accessed only de-identified data routinely collected as part of health screening examinations (IRB No: KBSMC2020-06-001). All methods were performed in accordance with relevant guidelines and regulations (the Declaration of Helsinki and comparable ethical principles).

## Measurements

All examinations were completed at the Kangbuk Samsung Hospital Total Healthcare Center in Seoul and Suwon. Data on demographic characteristics; working hours; smoking status; alcohol consumption; physical activity; education level; medication use; and medical history of hypertension, diabetes, and liver disease were collected by standardized, self-administered questionnaires at each visit [20]. On the day of the health examination, a trained nurse checked the questionnaire for blanks, and during the final stage of the health examination, a trained doctor double-checked whether there were any incorrect or blank marks on the questionnaire while conducting a face-to-face interview with the examinee. Smoking status was categorized as never, former, or current smokers. Alcohol consumption was categorized as ≥10 g/day and <10 g/day. The weekly frequency of moderate- or vigorous-intensity physical activity was also assessed and categorized as <3 or ≥3 times per week, respectively. Education level was categorized as less than college graduate or college graduate or more [23].

Working hours were identified using the following question: "How many hours did you work in a week on average in your job for the past year, including overtime?" According to the International Labour Organization, working >48 hours per week is considered a major job stress, and working >60 hours per week is associated with occurrence of cerebro-

cardiovascular diseases [24, 25]. In addition, the Labor Standards Act of Korea states that working hours of adults should not exceed 40 hours per week excluding recess hours (12 additional hours per week are allowed with workers' permission) and working hours of adolescents should not exceed 35 hours per week (5 additional hours per week are allowed with workers' permission) [26]. Based on this information, the weekly working hours on average in the past year were categorized as 35–40, 41–52, 53–60, and >60 hours per week.

Blood pressure, height, and weight were measured by trained nurses. Obesity was defined as body mass index (BMI) ≥25 kg/m$^2$. Hypertension was defined as a systolic blood pressure ≥140 mmHg, a diastolic blood pressure ≥90 mmHg, a self-reported history of hypertension, or current use of anti-hypertensive medications. Fasting blood measurements included glucose, total cholesterol, low-density lipoprotein cholesterol (LDL-C), triglycerides, high-density lipoprotein cholesterol (HDL-C), aspartate aminotransferase (AST), alanine aminotransferase (ALT), gamma-glutamyltransferase (GGT), and high-sensitivity C-reactive protein (hsCRP). DM was defined as a fasting serum glucose level of ≥126 mg/dL, a hemoglobin A1c level ≥6.5%, a self-reported history of DM, or current use of anti-diabetic medications. Insulin resistance was assessed with the homeostatic model assessment-insulin resistance (HOMA-IR) equation as follows: fasting insulin (μU/mL) × fasting glucose (mg/dL)/405 [5, 27].

Abdominal ultrasounds were performed using a Logic Q700 MR 3.5-MHz transducer (GE, Milwaukee, WI) by experienced radiologists who were unaware of the study aims. Images were obtained in a standard fashion with patients in the supine position with their right arm raised above their head [28]. An ultrasonographic diagnosis of fatty liver was defined as the presence of a diffuse increase of fine echoes in the liver parenchyma compared with the kidney or spleen parenchyma [29]. The inter- and intra-observer reliability in the diagnosis of fatty liver was very high (kappa statistics of 0.74 and 0.94, respectively) [30]. NAFLD was defined as the presence of fatty liver in the absence of excessive alcohol use (≥20 g/day for women and ≥30 g/day for men) or other identifiable cause, as described in the exclusion criteria [16]. Because we had already excluded participants with excessive alcohol use, as well as other identifiable causes of fatty liver at baseline as described in the exclusion criteria, incident cases of fatty liver were considered NAFLD.

## Statistical analysis

The chi-square test and one-way ANOVA were used to compare characteristics of study participants stratified by working hours at baseline. The primary endpoint was the development of incident NAFLD. Participants were followed from the baseline visit to the NAFLD diagnosis visit or to the last available visit before December 31, 2018, whichever came first. Incidence rates were calculated as the number of incident cases divided by person-years of follow-up.

Hazard ratios (HRs) and 95% confidence intervals (CIs) for incident NAFLD were estimated using Cox proportional hazards regression analyses. We initially adjusted for age, sex, center (Seoul and Suwon), and year of screening exam (Model 1). Model 2 was further adjusted for smoking status, alcohol intake, education level, history of diabetes, medication for diabetes, history of hypertension, medication for hypertension and medication for dyslipidemia. To explore the mechanism underlying the observed associations between working hours and NAFLD risk, Model 3 was further adjusted for BMI, hsCRP, and HOMA-IR. We assessed the proportional hazards assumption by examining graphs of estimated log (-log) survival; no violation of the assumption was found. To determine the linear trend of incidence, the number of categories was used as a continuous variable and tested in each model. To evaluate the effects of changes in covariates during follow-up, we conducted additional analyses using covariates as time-varying covariates in the models.

In addition, stratified analyses in predefined subgroups were performed by BMI ($<25$ versus $\geq 25$ kg/m$^2$), hsCRP ($<1.0$ versus $\geq 1.0$ mg/L), HOMA-IR ($<2.5$ versus $\geq 2.5$), and shift work (daytime work versus shift work). Interactions between working hours categories and subgroup characteristics were tested using likelihood ratio tests, which compared models with and without multiplicative interaction terms.

Statistical analyses were performed using STATA version 16.1 (StataCorp LP, College Station, TX). All reported P values were two-tailed. A P value $<0.05$ was considered statistically significant.

## Results

At baseline, the mean (standard deviation) age and BMI of 79,048 participants were 36.3 (7.0) years and 22.2 (2.6) kg/m$^2$, respectively (Table 1). Weekly working hours were positively associated with male sex, current smoking status, alcohol intake, regular exercise, education level,

**Table 1. Baseline characteristics of study participants stratified by weekly working hours.**

| Characteristics | Overall | Weekly working hours | | | | P-value for trend |
|---|---|---|---|---|---|---|
| | | 35–40 | 41–52 | 53–60 | >60 | |
| Number | 79,048 | 18,558 | 41,480 | 13,620 | 5,390 | |
| Age (years)* | 36.3 (7.0) | 38.1 (7.7) | 35.7 (6.8) | 35.8 (6.4) | 35.8 (6.7) | <0.001 |
| Male (%) | 58.8 | 37.9 | 62.3 | 71.1 | 73.0 | <0.001 |
| Current smoker (%) | 17.5 | 11.9 | 17.3 | 22.3 | 25.7 | <0.001 |
| Alcohol intake (%)[a] | 87.4 | 82.2 | 88.2 | 90.5 | 90.8 | <0.001 |
| Regular exercise (%)[b] | 12.2 | 13.0 | 12.3 | 11.3 | 11.2 | <0.001 |
| High education level (%)[c] | 87.7 | 82.4 | 88.5 | 91.5 | 89.9 | <0.001 |
| Hypertension (%) | 5.6 | 5.9 | 5.3 | 5.9 | 5.6 | 0.646 |
| Diabetes (%) | 1.06 | 1.37 | 0.97 | 0.97 | 0.89 | <0.001 |
| Medication for dyslipidemia (%) | 1.23 | 1.44 | 1.13 | 1.20 | 1.34 | 0.226 |
| Obesity (%)[d] | 14.5 | 12.5 | 14.5 | 16.2 | 16.9 | <0.001 |
| BMI (kg/m$^2$)* | 22.2 (2.6) | 21.9 (2.6) | 22.3 (2.6) | 22.5 (2.6) | 22.6 (2.6) | <0.001 |
| Systolic BP (mmHg)* | 105.4 (11.6) | 103.6 (11.9) | 105.7 (11.5) | 106.5 (11.3) | 106.8 (11.2) | <0.001 |
| Diastolic BP (mmHg)* | 67.9 (9.1) | 67.0 (9.3) | 68.0 (9.0) | 68.6 (8.9) | 68.8 (8.7) | <0.001 |
| Glucose (mg/dL)* | 92.0 (9.8) | 91.9 (10.7) | 92.0 (9.6) | 92.1 (9.8) | 92.2 (8.6) | 0.055 |
| Total cholesterol (mg/dL)* | 188.4 (31.5) | 187.6 (32.0) | 188.3 (31.4) | 189.4 (31.0) | 189.7 (31.2) | <0.001 |
| LDL-C (mg/dL)* | 115.6 (29.5) | 113.7 (29.7) | 115.7 (29.5) | 117.2 (29.3) | 117.3 (29.2) | <0.001 |
| HDL-C (mg/dL)* | 61.1 (14.7) | 63.0 (15.0) | 60.8 (14.6) | 59.8 (14.2) | 59.8 (14.2) | <0.001 |
| Triglycerides (md/dL)[#] | 79 (59–109) | 75 (56–104) | 79 (59–109) | 81 (60–112) | 82 (61–113) | <0.001 |
| AST (U/L)[#] | 18 (16–22) | 18 (15–21) | 18 (16–22) | 18 (16–22) | 19 (16–22) | <0.001 |
| ALT (U/L)[#] | 16 (12–21) | 14 (11–19) | 16 (12–22) | 17 (12–23) | 17 (13–23) | <0.001 |
| GGT (U/L)[#] | 17 (12–26) | 15 (11–23) | 18 (13–26) | 19 (14–27) | 19 (14–28) | <0.001 |
| hsCRP (mg/L)[#] | 0.03 (0.02–0.07) | 0.03 (0.02–0.07) | 0.03 (0.02–0.07) | 0.04 (0.02–0.07) | 0.04 (0.02–0.07) | 0.029 |
| HOMA-IR[#] | 1.08 (0.73–1.52) | 1.06 (0.72–1.52) | 1.08 (0.74–1.53) | 1.07 (0.73–1.51) | 1.08 (0.75–1.53) | 0.86 |

Data are expressed as

*mean (standard deviation)

[#]median (interquartile range), or percentage.

[a]$\geq$ 10g/day

[b]$\geq$ 3 times/week

[c]$\geq$ College graduate

[d]BMI $\geq$ 25kg/m$^2$.

**Table 2. Development of NAFLD according to weekly working hours.**

| Weekly working hours | Person-years (PY) | Incident cases | Incidence density (per $10^2$ PY) (95% CI) | Multivariable-adjusted HR (95% CI)[a] | | | HR (95% CI)[b] in model using time-dependent variables |
|---|---|---|---|---|---|---|---|
| | | | | Model 1[*] | Model 2[**] | Model 3[***] | |
| 35–40 | 61,646.0 | 2,691 | 4.37 (4.20–4.53) | 1.00 (reference) | 1.00 (reference) | 1.00 (reference) | 1.00 (reference) |
| 41–52 | 143,813.3 | 8,150 | 5.67 (5.55–5.79) | 1.02 (0.98–1.07) | 1.03 (0.98–1.08) | 1.07 (1.02–1.13) | 1.06 (1.01–1.12) |
| 53–60 | 47,977.8 | 3,016 | 6.29 (6.07–6.51) | 1.04 (0.99–1.10) | 1.05 (0.99–1.11) | 1.06 (1.00–1.13) | 1.05 (0.99–1.12) |
| >60 | 18,531.2 | 1,238 | 6.68 (6.32–7.06) | 1.09 (1.02–1.17) | 1.09 (1.01–1.17) | 1.13 (1.05–1.23) | 1.14 (1.06–1.23) |
| P for trend | | | | 0.012 | 0.013 | 0.009 | 0.006 |

[a] Estimated from Cox proportional hazard models.

[b] Estimated from Cox proportional hazard models with alcohol intake, smoking status, regular exercise, BMI, hsCRP, and HOMA-IR as time-dependent variables and baseline age, sex, center, year of screening exam, education level, weekly working hours, history of diabetes, medication for diabetes, history of hypertension, medication for hypertension, and medication for dyslipidemia as time-fixed variables.

[*] Model 1 was adjusted for age, sex, center, and year of screening examination.

[**] Model 2: model 1 plus adjustment for smoking status, alcohol intake, regular exercise, education level, history of diabetes, medication for diabetes, history of hypertension, medication for hypertension, and medication for dyslipidemia.

[***] Model 3: model 2 plus adjustment for BMI, hsCRP, and HOMA-IR.

obesity, BMI, systolic and diastolic BP, total cholesterol, LDL-C, triglycerides, AST, ALT, and GGT, whereas they were inversely associated with age, diabetes, and HDL-C.

Table 2 shows the relationship between weekly working hours and the incidence of NAFLD. Over 271,968.3 person-years of follow-up (median follow-up, 6.6 years; interquartile range, 4.9–6.9 years), 15,095 participants developed NAFLD (incidence density, 5.55 per 100 person-years). Participants with longer working hours had a higher incidence of NAFLD. Across all models, working >60 hours were associated with a significantly higher risk of incident NAFLD than working 35–40 hours. Especially, in Model 3, multivariable-adjusted HRs (95% CI) of incident NAFLD for working 41–52 hours, 53–60 hours and, >60 hours to compared with working 35–40 hours were 1.07 (1.02–1.13), 1.06 (1.00–1.13), and 1.13 (1.05–1.23), respectively. Even after introducing confounders (alcohol intake, smoking status, regular exercise, BMI, hsCRP, and HOMA-IR) as time-varying covariates, the association between working hours and NAFLD was still observed in the time-dependent model.

In subgroup analyses (Table 3), the association between working hours and incident NAFLD was consistently observed in those with hsCRP <1.0 mg/L (versus ≥1.0 mg/L) and daytime work (versus shift work). The association was similar across subgroups stratified by BMI (<25 versus ≥25 kg/m$^2$) and HOMA-IR (<2.5 versus ≥2.5). There was no significant interaction with predetermined subgroups.

## Discussion

A previous cross-sectional study of 194,625 Koreans showed that the prevalence of NAFLD tended to increase with increasing working hours, especially in participants with over 52 hours per week; the prevalence of NALFD differed according to sleep duration as well as working hours, suggesting that increased working hours would lead to decreased sleeping hours, reduced physical activity, and increased obesity incidence [18]. In addition, two Chinese cross-sectional studies found an association between rotating night shift work and the prevalence of NAFLD, which might be due to circadian disruption [19, 31]. However, several other

**Table 3. Hazard ratios[a] (95% CI) for NAFLD by weekly working hours in clinically relevant subgroups.**

| | Weekly working hours | | | | | |
|---|---|---|---|---|---|---|
| Subgroups | 35–40 | 41–52 | 53–60 | >60 | P for trend | P for interaction |
| BMI | | | | | | 0.423 |
| <25 kg/m$^2$ (n = 67,466) | 1.00 (reference) | 1.08 (1.01–1.15) | 1.06 (0.98–1.14) | 1.12 (1.02–1.23) | 0.076 | |
| ≥25 kg/m$^2$ (n = 11,481) | 1.00 (reference) | 1.08 (0.98–1.19) | 1.09 (0.98–1.22) | 1.20 (1.04–1.37) | 0.021 | |
| hsCRP | | | | | | 0.83 |
| <1.0 mg/L (n = 64,779) | 1.00 (reference) | 1.08 (1.02–1.14) | 1.07 (1.00–1.14) | 1.14 (1.05–1.23) | 0.008 | |
| ≥1.0 mg/L (n = 667) | 1.00 (reference) | 0.88 (0.52–1.48) | 0.81 (0.45–1.46) | 1.11 (0.53–2.33) | 0.959 | |
| HOMA-IR | | | | | | 0.182 |
| <2.5 (n = 75,028) | 1.00 (reference) | 1.07 (1.02–1.14) | 1.06 (0.99–1.13) | 1.12 (1.03–1.21) | 0.034 | |
| ≥2.5 (n = 3,739) | 1.00 (reference) | 1.09 (0.92–1.29) | 1.28 (1.05–1.56) | 1.45 (1.12–1.87) | 0.001 | |
| Shift work[b] | | | | | | 0.414 |
| Daytime work (n = 71,303) | 1.00 (reference) | 1.08 (1.02–1.14) | 1.07 (1.00–1.14) | 1.15 (1.06–1.25) | 0.006 | |
| Shift work (n = 7,117) | 1.00 (reference) | 1.09 (0.93–1.28) | 1.04 (0.85–1.28) | 1.01 (0.78–1.29) | 0.971 | |

[a] Estimated from Cox proportional hazard models adjusted for age, sex, center, year of screening examination, smoking status, alcohol intake, regular exercise, education level, history of diabetes, medication for diabetes, history of hypertension, medication for hypertension, medication for dyslipidemia, BMI, hsCRP, and HOMA-IR.

[b] In the question "In the past year, during which time of the day did you work the most?", daytime work was defined as participants who answered that "I worked mostly during the day (between 6 AM and 6 PM)," and shift work was defined as participants who answered that "I worked during other hours."

studies showed no association between shift work and NAFLD. Therefore, the relationship between shift work and NAFLD is still controversial [32, 33].

To our knowledge, most of the existing studies on the relationship between work-related factors and NAFLD are only cross-sectional studies; thus, we conducted a longitudinal cohort study to elucidate the causal relationship. Our large-scale cohort study, in which participants had no fatty liver at baseline, is the largest study to analyze the causality between working hours and NAFLD. Working >60 hours based on self-reports was significantly associated with an increased risk of incident NAFLD compared with working 35–40 hours. Increased baseline working hours had a dose-response relationship with the incidence of NAFLD. Moreover, the association was consistently observed when changes in potential confounders during follow-up were treated as time-varying covariates.

The mechanisms by which working hours affects the development of NAFLD are not fully understood. A previous cohort study showed that metabolically healthy individuals with no metabolic abnormalities had a higher risk of NAFLD when they were overweight or obese [20]. Consistently, in order to explore whether the increased risk of NAFLD associated with long working hours was mediated by BMI, we performed a stratified analysis adjusted for obesity (BMI <25 versus ≥25 kg/m$^2$) and found that the association of long working hours with development of NAFLD remained significant in both subgroups with no interaction. In addition, another cohort study showed that relatively higher hsCRP levels increased the risk of developing NAFLD [34]. Such association can be explained by chronic low-grade inflammation due to oxidative stress, which is one of the major mechanisms of NAFLD. Therefore, we also performed a stratified analysis in subgroups by hsCRP (<1.0 versus ≥1.0 mg/L) and found that the association still remained significant in those with hsCRP <1.0 mg/L. Because of the exceedingly small number of participants with hsCRP ≥1.0 mg/L and possible healthy worker effects, HRs became insignificant even though the HRs in working >60 hours were similar in both subgroups of hsCRP. Moreover, long working hours were significantly related to psychosocial stress [35], and stress activates the hypothalamic-pituitary-adrenal axis, which contribute to insulin resistance by affecting the release of

cortisol [36]. From the point of view of such a relationship, long working hours may be associated with insulin resistance. Notably, insulin resistance is closely related to NAFLD and is known as one of the main mechanisms of the progression of NAFLD [37]. Therefore, we performed a stratified analysis in subgroups by HOMA-IR ($<$2.5 versus $\geq$2.5). The association still remained significant in both subgroups with no interaction and was even higher in those with HOMA-IR $\geq$2.5 than in those with HOMA-IR $<$2.5. Taken together, these findings suggest that long working hours affect the development of NAFLD through various underlying mechanisms. Still, the effect of long working hours on the liver is not yet clear, and further studies are needed to elucidate the mechanisms underlying these direct associations.

Some limitations should be noted in the present study. First, NAFLD was diagnosed with abdominal ultrasound rather than more accurate methods such as liver biopsy. However, liver biopsy is an invasive procedure that can be accompanied by complications. In large-scale epidemiological studies, ultrasound is widely used due to its accuracy for detecting fatty liver [38]. Second, the work-related variables and lifestyle variables were collected through a self-administered structured questionnaire. Therefore, measurement errors from those variables could not be excluded. However, since there was no reason for the examinee to underreport or overreport out of self-interest, it was considered that the results would not be significantly affected by differential misclassification. Third, working hours might change during the follow-up period, which could affect the study results. During the entire follow-up period, the percentage of participants whose group of working hours moved only once or less to one adjacent group was approximately 70% of the total population (data not shown). Consequently, if all participants with small fluctuations in working hours were excluded, selection bias could occur. Furthermore, if the same analysis is performed on 30,311 participants without changes in working hours, the risk is much stronger when it exceeds 60 hours (HR = 2.90, 95% CI 2.47–3.41, S1 Table), suggesting that our study results could be underestimated. Fourth, we could not adjust for sleep duration and diet, which could be risk factors for NAFLD. Therefore, further research is needed to explicate the exact mechanism. Lastly, our study participants were young and middle-aged Korean with relatively good health status and high educational level. Thus, our findings may not be generalized to other populations.

Despite the limitations, our study has several notable strengths. This longitudinal study conducted in a large sample size with a relatively large time of follow-up is the first to investigate the temporal association of long working hours with the risk of incident NAFLD. Our findings obtained in the relatively healthy young and middle-aged population are less likely to be affected by survivor bias from comorbidities or the use of multiple medications.

In conclusion, our large-scale cohort study of young and middle-aged individuals demonstrated a causal relationship between long working hours and the incidence of NAFLD. Our findings suggest that long working hours are a risk factor for NAFLD. Further studies are needed to elucidate the potential mechanisms underlying this association.

## Supporting information

**S1 Table. Development of NAFLD according to weekly working hours among participants with no changes of working hours (n = 30,311).**
(DOCX)

## Acknowledgments

This study was conducted based on the data provided by Kangbuk Samsung Health Study. The authors thank all study participants and the study personnel for their dedication and continuing support.

## Author Contributions

**Conceptualization:** Yesung Lee, Woncheol Lee.

**Formal analysis:** Yesung Lee, Woncheol Lee.

**Investigation:** Yesung Lee, Woncheol Lee.

**Methodology:** Yesung Lee, Woncheol Lee.

**Writing – original draft:** Yesung Lee.

**Writing – review & editing:** Eunchan Mun, Soyoung Park, Woncheol Lee.

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
