## [Decision Letter · Decision Letter 0]

20 May 2021

PONE-D-21-10560

Long working hours are associated with a higher risk of non-alcoholic fatty liver disease: a large population-based Korean cohort study

PLOS ONE

Dear Dr. Lee,

Thank you for submitting your manuscript to PLOS ONE. After careful consideration, we feel that it has merit but does not fully meet PLOS ONE’s publication criteria as it currently stands. Therefore, we invite you to submit a revised version of the manuscript that addresses the points raised during the review process.

We look forward to receiving your revised manuscript.

Kind regards,

Jee-Fu Huang, M.D., Ph.D.

Academic Editor

PLOS ONE

Journal Requirements:

Reviewers' comments:

Reviewer's Responses to Questions

**Comments to the Author**

1. Is the manuscript technically sound, and do the data support the conclusions?

Reviewer #1: Yes

Reviewer #2: Yes

2. Has the statistical analysis been performed appropriately and rigorously? 

Reviewer #1: Yes

Reviewer #2: Yes

3. Have the authors made all data underlying the findings in their manuscript fully available?

Reviewer #1: Yes

Reviewer #2: Yes

4. Is the manuscript presented in an intelligible fashion and written in standard English?

Reviewer #1: Yes

Reviewer #2: Yes

5. Review Comments to the Author

Reviewer #1: This study aims to evaluate the association between working hours and the development of NAFLD. 79,048 Korean adults without NAFLD at baseline were included in the retrospective study and weekly working hours were categorized into 35–40, 41–52, 53–60, and >60 hours. The median follow-up period was 6.6 years and 15,095 participants developed new-onset NAFLD. After adjustment for confounders, the hazard ratios for the development of NAFLD in 41–52, 53–60, and >60 working hours compared with that in 35–40 working hours were 1.07 (1.02–1.13), 1.06 (1.00–1.13), and 1.13 (1.05–1.23), respectively. They concluded that long working hours are a risk factor for NAFLD.

Comments are as follows:

1. Patients were categorized into four groups according to working hours at baseline. Since the median follow-up was 6.6 years, working hours might change during follow-up period. Will the change of working hours affect the outcome and how to prevent this bias?

2. The authors categorized weekly working hours as 35–40, 41–52, 53–60, and >60 hours based on the policy of Labor Standards Act of Korea.

There are questions about this classification.

a. Why were subjects with weekly working hours <35 excluded from this study?

b. The group of working hours between 35 and 40 was based on the policy for adolescents. Because subjects with age <19 years were excluded from this study, why did the authors include the group of working hours between 35 and 40?

c. Is it more reasonable to simply adapt the suggestion of International Labour Organization (48 hours and 60 hours) as mentioned by the authors in the Measurements section?

3. In MATERIALS AND METHODS section, alcohol consumption was divided into ≥10 g/day and <10 g/day. Since alcohol intake of ≥30 g/day for men or ≥20 g/day for women has been excluded from this study. What is the clinical significance of this classification?

Reviewer #2: Non-alcoholic fatty liver disease (NAFLD) is an increasingly cause of chronic liver disease. Several factors are associated with the development of NAFLD, such as obesity, type 2 diabetes mellitus. In this study, Lee et al. investigated the association between working hours and the development of NAFLD. They found that long working hours, were independently associated with incident NAFLD. Although the results were clinically interesting, several points need be critically addressed.

1. The time of baseline data collection is not clear. The authors should describe when the baseline data was collected in this study.

2. In this study, alcohol consumption was categorized as ≥10 g/day and <10 g/day. Please describe the reason for this category.

3. In this study, obesity was defined as body mass index (BMI) ≥25 kg/m2. However, the cutoff for obesity is 23.0 kg/m2 in Asia-Pacific countries, including Korea. The authors may perform subgroups analysis by BMI <23 versus ≥23 kg/m2.

4. The association between the change in body weight or BMI during the follow-up period and development of NAFLD should be taken into consideration.

5. The authors should describe how to evaluate the reliability and validity of the self-administered questionnaires.

---

## [Author Response · Author response to Decision Letter 0]

25 Jun 2021

Reviewer #1: This study aims to evaluate the association between working hours and the development of NAFLD. 79,048 Korean adults without NAFLD at baseline were included in the retrospective study and weekly working hours were categorized into 35–40, 41–52, 53–60, and >60 hours. The median follow-up period was 6.6 years and 15,095 participants developed new-onset NAFLD. After adjustment for confounders, the hazard ratios for the development of NAFLD in 41–52, 53–60, and >60 working hours compared with that in 35–40 working hours were 1.07 (1.02–1.13), 1.06 (1.00–1.13), and 1.13 (1.05–1.23), respectively. They concluded that long working hours are a risk factor for NAFLD.

Comments are as follows:

1. Patients were categorized into four groups according to working hours at baseline. Since the median follow-up was 6.6 years, working hours might change during follow-up period. Will the change of working hours affect the outcome and how to prevent this bias?

Response: We thank the reviewer for the opportunity to clarify this issue. During the entire follow-up period, the percentage of participants with unchanged working hours was 38.35%, and the percentage of participants whose group of working hours moved only once or less to one adjacent group was about 70% of the total population. Therefore, we can assume that there is little variation in working hours between the participants of this study. Furthermore, when the same analysis is performed on 30,311 people who have no change in working hours, the risk is much stronger when it exceeds 60 hours (HR = 2.90, 95% CI 2.47–3.41, S1 Table). However, if all the people with small fluctuations in working hours are excluded, selection bias may occur. Therefore, the current method seems reasonable, and the results are considered to be underestimations. Besides pointing out that the change in working hours is a limitation of our study, we added that the results could be underestimated in the limitation paragraph of the DISCUSSION (Lines 205–211) along with S1 Table. 

S1 Table. Development of NAFLD according to weekly working hours among participants with no changes in working hours (n=30,311)

Weekly working hours Person-years (PY) Incident cases Incidence density 

(per 102 PY) 

(95% CI) Multivariable-adjusted HR (95% CI)a HR (95% CI)b in model using time-dependent variables

 Model 1* Model 2** Model 3*** 

35-40 33185.3 1532 4.62 (4.39-4.85) 1.00 (reference) 1.00 (reference) 1.00 (reference) 1.00 (reference)

41-52 50238.2 3792 7.55 (7.31-7.79) 1.20 (1.12-1.28) 1.20 (1.12-1.28) 1.21 (1.12-1.31) 1.19 (1.10-1.29)

53-60 3604.2 552 15.32 (14.09-16.65) 2.54 (2.30-2.82) 2.47 (2.23-2.74) 2.54 (2.26-2.85) 2.57 (2.29-2.89)

>60 1499.3 237 15.80 (13.92-17.95) 2.73 (2.38-3.15) 2.69 (2.33-3.11) 2.90 (2.47-3.41) 3.00 (2.56-3.52)

P for trend <0.001 <0.001 <0.001 <0.001

a Estimated from Cox proportional hazard models.

b Estimated from Cox proportional hazard models with alcohol intake, smoking status, regular exercise, BMI, hsCRP and HOMA-IR as time-dependent variables and baseline age, sex, center, year of screening exam, education level, weekly working hours, history of diabetes, medication for diabetes, history of hypertension, medication for hypertension and medication for dyslipidemia as time-fixed variables

* Model 1 was adjusted for age, sex, center, and year of screening examination.

** Model 2: model 1 plus adjustment for smoking status, alcohol intake, regular exercise, education level, history of diabetes, medication for diabetes, history of hypertension, medication for hypertension and medication for dyslipidemia.

*** Model 3: model 2 plus adjustment for BMI, hsCRP and HOMA-IR

2. The authors categorized weekly working hours as 35–40, 41–52, 53–60, and >60 hours based on the policy of Labor Standards Act of Korea.

There are questions about this classification.

a. Why were subjects with weekly working hours <35 excluded from this study?

Response: We thank the reviewer for this comment. In our study, part-time and full-time workers were not classified separately. However, to completely understand the effect of long working hours, only participants who thought they were full-time workers were included in the analysis. According to the ILO report, a person working less than 35 hours was defined as a part-time worker (Lee SH, et al. Working Time Around the World: Trends in Working Hours, Laws and Policies in a Global Comparative Perspective. Oxford: Routledge; 2007.).

b. The group of working hours between 35 and 40 was based on the policy for adolescents. Because subjects with age <19 years were excluded from this study, why did the authors include the group of working hours between 35 and 40?

Response: We thank the reviewer for the pertinent comment. There were fewer than 10 participants aged under 19 years in the entire population of our study, and we wanted to exclude the effects of adolescents. The reason for using weekly working 35-40 hours as a reference was to understand the health effects of long working hours among full-time workers. Since this study is conducted among Korean workers, the legal maximum working hours for adolescents in Korean law was used as a reference. In addition, several previous studies of long working hours have used 35–40 hours as the standard working hours for workers who worked more than 35 hours (Kim W, et al. Effect of working hours and precarious employment on depressive symptoms in South Korean employees: a longitudinal study. Occup Environ Med. 2016;73(12):816-22; Kivimäki M, et al. Long working hours, socioeconomic status, and the risk of incident type 2 diabetes: a meta-analysis of published and unpublished data from 222 120 individuals. The Lancet Diabetes & Endocrinology. 2015;3(1):27-34.).

c. Is it more reasonable to simply adapt the suggestion of International Labour Organization (48 hours and 60 hours) as mentioned by the authors in the Measurements section?

Response: We thank the reviewer for this important comment. We defined the classification of working hours based on the fact that Korea has one of the countries with the highest annual working hours per worker among OECD countries and that the participants of this study were all Koreans who are legally allowed to work up to 52 hours (except for some occupations such as medical personnel or transporter). However, when the sensitivity analysis was performed according to the ILO classification, the risk of NAFLD still significantly increased with respect to long working hours (>60 hours: HR = 1.13, 95% CI 1.05–1.23, Table 1 not shown in the manuscript), showing no difference from our results.

Table 1. Development of NAFLD according to weekly working hours in the ILO classification

Weekly working hours Person-years (PY) Incident cases Incidense density 

(per 102 PY) 

(95% CI) Multivariable-adjusted HR (95% CI)a HR (95% CI)b in model using time-dependent variables

 Model 1* Model 2** Model 3*** 

35-40 61646.0 2691 4.37 (4.20-4.53) 1.00 (reference) 1.00 (reference) 1.00 (reference) 1.00 (reference)

41-48 64951.5 3369 5.19 (5.01-5.37) 1.03 (0.98-1.09) 1.03 (0.98-1.09) 1.08 (1.01-1.14) 1.06 (1.00-1.13)

49-60 126839.6 7797 6.15 (6.01-6.29) 1.02 (0.98-1.07) 1.04 (0.99-1.09) 1.07 (1.01-1.13) 1.06 (1.00-1.11)

>60 18531.2 1238 6.68 (6.32-7.06) 1.09 (1.02-1.17) 1.09 (1.02-1.17) 1.13 (1.05-1.23) 1.14 (1.06-1.23)

P for trend 0.093 0.035 0.005 0.008

a Estimated from Cox proportional hazard models.

b Estimated from Cox proportional hazard models with alcohol intake, smoking status, regular exercise, BMI, hsCRP and HOMA-IR as time-dependent variables and baseline age, sex, center, year of screening exam, education level, weekly working hours, history of diabetes, medication for diabetes, history of hypertension, medication for hypertension and medication for dyslipidemia as time-fixed variables

* Model 1 was adjusted for age, sex, center, and year of screening examination.

** Model 2: model 1 plus adjustment for smoking status, alcohol intake, regular exercise, education level, history of diabetes, medication for diabetes, history of hypertension, medication for hypertension and medication for dyslipidemia.

*** Model 3: model 2 plus adjustment for BMI, hsCRP and HOMA-IR

3. In MATERIALS AND METHODS section, alcohol consumption was divided into ≥10 g/day and <10 g/day. Since alcohol intake of ≥30 g/day for men or ≥20 g/day for women has been excluded from this study. What is the clinical significance of this classification?

Response: We thank the reviewer for this thoughtful comment. This classification was defined considering the cultural aspects of drinking in Korea, where alcohol consumption is high and it is difficult to completely avoid alcohol. The group that consumed relatively high amounts of alcohol and the group that did not consume alcohol were classified by one standard drink used in Korea (10 grams of pure alcohol based on the WHO definition). We referred to previous studies on NAFLD, which set one standard drink as the criteria for alcohol consumption (Jung HS, et al. Smoking and the Risk of Non-Alcoholic Fatty Liver Disease: A Cohort Study. Am J Gastroenterol. 2019;114(3):453-63.) (Kim Y, et al. Metabolically healthy versus unhealthy obesity and risk of fibrosis progression in non-alcoholic fatty liver disease. Liver Int. 2019;39(10):1884-94.).

 

Reviewer #2: Non-alcoholic fatty liver disease (NAFLD) is an increasingly cause of chronic liver disease. Several factors are associated with the development of NAFLD, such as obesity, type 2 diabetes mellitus. In this study, Lee et al. investigated the association between working hours and the development of NAFLD. They found that long working hours, were independently associated with incident NAFLD. Although the results were clinically interesting, several points need be critically addressed.

1. The time of baseline data collection is not clear. The authors should describe when the baseline data was collected in this study.

Response: We thank the reviewer for this comment. We described that the study included the participants who underwent health examinations from January 1, 2012, to December 31, 2017 in Lines 64–65. To avoid misunderstanding, we added the following explanation to the Study population section (Lines 66–67): “During this period, the baseline data were collected at the time of the first health examination visit.”

2. In this study, alcohol consumption was categorized as ≥10 g/day and <10 g/day. Please describe the reason for this category

Response: We thank the reviewer for the pertinent comment. The group that consumed relatively high amounts of alcohol and the group that did not consume alcohol were classified by one standard drink used in Korea (10 grams of pure alcohol based on the WHO definition). We referred to previous studies on NAFLD, which set one standard drink as the criteria for alcohol consumption (Jung HS, et al. Smoking and the Risk of Non-Alcoholic Fatty Liver Disease: A Cohort Study. Am J Gastroenterol. 2019;114(3):453-63.) (Kim Y, et al. Metabolically healthy versus unhealthy obesity and risk of fibrosis progression in non-alcoholic fatty liver disease. Liver Int. 2019;39(10):1884-94.).

3. In this study, obesity was defined as body mass index (BMI) ≥25 kg/m2. However, the cutoff for obesity is 23.0 kg/m2 in Asia-Pacific countries, including Korea. The authors may perform subgroups analysis by BMI <23 versus ≥23 kg/m2.

Response: We thank the reviewer for the in-depth comment. In Korea, the cut-off value for obesity is 25 kg/m2, as recommended by the Korean Society for the Study of Obesity. This is derived from the Asia-Pacific classification of obesity established in 2000 by the WHO, where 23 kg/m2 is the cut-off value for being overweight (WHO. The Asia-Pacific perspective: Redefining obesity and its treatment. Sydney: Health Communications Australia; 2000). The Korean Society for the Study of Obesity has defined the cut-off value for obesity as 25 kg/m2 because the prevalence and mortality of obesity-related diseases increase when the BMI exceeds 25 kg/m2 in Asians (Korean Society for the Study of Obesity. Guideline for the management of obesity 2020. Seoul: Korean Society for the Study of Obesity; 2020). Furthermore, many Korean studies on NAFLD have set the criteria for obesity as 25 kg/m2 (Chang Y, et al. Nonheavy Drinking and Worsening of Noninvasive Fibrosis Markers in Nonalcoholic Fatty Liver Disease: A Cohort Study. Hepatology. 2019;69(1):64-75.). When subgroup analysis was performed based on overweight status, the HR increased in long working hours with no significance (HR = 1.14, 95% CI 1.00–1.30, Table 2 not shown in the manuscript), but no interaction was observed. The hypothesis for this result is that people who maintain BMI at less than 23 kg/m2 did not show a high risk of NAFLD in long working hours because they are generally interested in health care and diet control. The limitation of our study is that there are no data on diet, which was mentioned in the limitation paragraph of the DISCUSSION. (Lines 211–213)

Table 2. Hazard ratiosa (95% CI) for NAFLD by weekly working hours subgrouped by overweight

　 Weekly working hours 　 　

Subgroup 35-40 41-52 53-60 >60 P for trend P for interaction

BMI 0.062

<23 kg/m2 (n=49,208) 1.00 (reference) 1.07 (0.98-1.17) 1.07 (0.96-1.19) 1.14 (1.00-1.30) 0.097 

≥23 kg/m2 (n=29,739) 1.00 (reference) 1.09 (1.02-1.16) 1.08 (1.00-1.17) 1.16 (1.05-1.28) 0.01 

a Estimated from Cox proportional hazard models adjusted for age, sex, center, year of screening examination, smoking status, alcohol intake, regular exercise, education level, history of diabetes, medication for diabetes, history of hypertension, medication for hypertension, medication for dyslipidemia, BMI, hsCRP and HOMA-IR

4. The association between the change in body weight or BMI during the follow-up period and development of NAFLD should be taken into consideration.

Response: We thank the reviewer for this important comment. During the entire follow-up period, the percentage of participants in the unchanged category of obesity was 87.82%, and the percentage of participants whose obesity status changed less than once was 97% of the total population. Therefore, it can be considered that there was almost no variation in obesity in participants of this study. When subgroup analysis was performed based on changes of obesity status, the HR of the changed group increased in long working hours with no significance (HR = 1.13, 95% CI 0.95–1.33, Table 3 not shown in the manuscript), but no interaction was observed. 

 

Table 3. Hazard ratiosa (95% CI) for NAFLD by weekly working hours subgrouped by changes of obesity

　 Weekly working hours 　 　

Subgroup 35-40 41-52 53-60 >60 P for trend P for interaction

changes of obesity 0.779

change (n=9,626) 1.00 (reference) 1.05 (0.94-1.17) 1.11 (0.97-1.26) 1.13 (0.95-1.33) 0.074 

no change (n=69,422) 1.00 (reference) 1.07 (1.01-1.14) 1.05 (0.98-1.13) 1.13 (1.03-1.23) 0.044 

a Estimated from Cox proportional hazard models adjusted for age, sex, center, year of screening examination, smoking status, alcohol intake, regular exercise, education level, history of diabetes, medication for diabetes, history of hypertension, medication for hypertension, medication for dyslipidemia, BMI, hsCRP and HOMA-IR

5. The authors should describe how to evaluate the reliability and validity of the self-administered questionnaires.

Response: We thank the reviewer for this thoughtful comment. We secured the reliability of the self-questionnaire by adding the following explanation to the measurements paragraph of MATERIALS AND METHODS in Lines 86–89: “On the day of the health examination, a trained nurse checked the questionnaire for blanks, and during the final stage of the health examination, a trained doctor double-checked whether there were any incorrect or blank marks on the questionnaire while conducting a face-to-face interview with the examinee.” In addition, when filling out a questionnaire of the examinee's medical history or working hours, there was no apparent benefit to the examinee. Therefore, since there was no reason for the examinee to underreport or overreport, we considered that the results would not be significantly affected by differential misclassification. This description was added to the limitation paragraph of the DISCUSSION (Lines 204–205).

---

## [Decision Letter · Decision Letter 1]

12 Jul 2021

Long working hours are associated with a higher risk of non-alcoholic fatty liver disease: a large population-based Korean cohort study

PONE-D-21-10560R1

Dear Dr. Lee,

We’re pleased to inform you that your manuscript has been judged scientifically suitable for publication and will be formally accepted for publication once it meets all outstanding technical requirements.

Kind regards,

Jee-Fu Huang, M.D., Ph.D.

Academic Editor

PLOS ONE

Additional Editor Comments (optional):

Reviewers' comments:

Reviewer's Responses to Questions

**Comments to the Author**

1. If the authors have adequately addressed your comments raised in a previous round of review and you feel that this manuscript is now acceptable for publication, you may indicate that here to bypass the “Comments to the Author” section, enter your conflict of interest statement in the “Confidential to Editor” section, and submit your "Accept" recommendation.

Reviewer #1: All comments have been addressed

Reviewer #2: All comments have been addressed

2. Is the manuscript technically sound, and do the data support the conclusions?

Reviewer #1: Yes

Reviewer #2: Yes

3. Has the statistical analysis been performed appropriately and rigorously? 

Reviewer #1: Yes

Reviewer #2: (No Response)

4. Have the authors made all data underlying the findings in their manuscript fully available?

Reviewer #1: Yes

Reviewer #2: Yes

5. Is the manuscript presented in an intelligible fashion and written in standard English?

Reviewer #1: Yes

Reviewer #2: Yes

6. Review Comments to the Author

Reviewer #1: The authors have answered all the questions and added the explanation in the manuscript. I have no further comments.

Reviewer #2: This revised manuscript is much improved and all previous comments were responded on point-to-point basis. I have no additional comments.

7. PLOS authors have the option to publish the peer review history of their article (what does this mean?). If published, this will include your full peer review and any attached files.

Reviewer #1: No

Reviewer #2: No

---

## [Editor Report · Acceptance letter]

14 Jul 2021

PONE-D-21-10560R1 

Long working hours are associated with a higher risk of non-alcoholic fatty liver disease: a large population-based Korean cohort study 

Dear Dr. Lee:

I'm pleased to inform you that your manuscript has been deemed suitable for publication in PLOS ONE. Congratulations! Your manuscript is now with our production department. 

Kind regards, 

on behalf of

Dr. Jee-Fu Huang 

Academic Editor

PLOS ONE